# Maturation process and international accreditation of trauma system in a rapidly developing country

**Ayman El-Menyar**[1,2]*, **Ahammad Mekkodathil**[2], **Mohammad Asim**[2], **Rafael Consunji**[3], **Gustav Strandvik**[4], **Ruben Peralta**[4], **Sandro Rizoli**[4], **Husham Abdelrahman**[4], **Monira Mollazehi**[5], **Ashok Parchani**[4], **Hassan Al-Thani**[4]

**1** Clinical Medicine, Weill Cornell Medical College, Doha, Qatar, **2** Clinical Research, Trauma Surgery, Hamad General Hospital, Doha, Qatar, **3** Injury Prevention Program, Trauma Surgery, Hamad General Hospital, Doha, Qatar, **4** Trauma Surgery, Hamad General Hospital, Doha, Qatar, **5** Qatar Trauma Registry, Hamad General Hospital, Doha, Qatar

* aymanco65@yahoo.com

**Data Availability Statement:** All relevant data are given within this manuscript.

**Funding:** The author(s) received no specific funding for this work.

## Abstract

### Background

As trauma systems mature, they are expected to improve patient care, reduce in-hospital complications and optimize outcomes. Qatar has a single trauma center, at the Hamad General Hospital, which serves as the hub for the trauma system that was verified as a level 1 trauma system by the Accreditation Canada International Distinction program in 2014. We hypothesized that this international accreditation was a major step, in the maturation process of the Qatar trauma system, that has positively impacted patient care, reduced complications and improved outcomes of trauma patients in such a rapidly developing country.

### Methods

A retrospective analysis of data was conducted for all trauma patients who were admitted between 2010 and 2018. Data were obtained from the level 1 trauma center registry at Hamad Medical Corporation. Patients were divided into Group 1- pre-accreditation (admitted from January 2010 to October 2014) and Group 2- post-accreditation (admitted from November 2014 to December 2018). Patients' characteristics and in-hospital outcomes were analyzed and compared. Data included patients' demographics; injury types, mechanism and injury severity scores, interventions, hospital stay, complications and mortality (pre-hospital and in-hospital). Time series analysis for mortality was performed using expert modeler.

### Results

Data from a total of 15,864 patients was collected and analyzed. Group 2 patients had more severe injuries in comparison to Group 1 (p<0.05). However, Group 2, had a lower complication rate (ventilator associated pneumonia (VAP)) and a shorter mean hospital length of stay (p<0.05). The overall mortality was 8%. In Group 2; the pre-hospital mortality was higher

**Competing interests:** The authors have declared that no competing interests exist

(52% vs. 41%, p = 0.001), while in-hospital mortality was lower (48% vs. 59%) compared to Group 1 (p = 0.001).

## Conclusions

The international recognition and accreditation of the trauma center in 2014 was the key factor in the maturation of the trauma system that improved the in-hospital outcomes. Accreditation also brought other benefits including a reduction in VAP and hospital length of stay. However, further studies are required to explore the maturation process of all individual components of the trauma system including the prehospital setting.

## Introduction

Traumatic injuries are one of the leading causes of death worldwide. In 2016, almost 4.9 million people lost their lives following unintentional or intentional injuries [1]. The WHO data showed that road traffic injuries (RTIs) were major contributors to fatal trauma following unintentional injuries whereas self-harm was the primary mechanism of intentional injuries [1]. For each trauma death, hundreds of people may sustain non-fatal injuries, disabilities and life-long health consequences of trauma. In addition to the morbidity and mortality burden, traumatic injuries are also associated with significant economic burden. The annual treatment cost of adult major trauma in the USA was estimated at USD 27 billion [2]. Although the ultimate goal is to prevent injuries from happening, public education, and providing high quality trauma care for patients with intentional and unintentional injuries remain crucial to prevent fatalities and reduce short- and long-term disabilities. An ideal trauma system was described by the Health Services Research Administration in 1992 as a continuing of care, emphasizing smooth management of/and transitions between recognition, stabilization, transport, treatment, rehabilitation and return of ideal functioning of the injured [3]. Evidence suggests that maturation of a trauma system improves patient care processes and reduces in-hospital complications, hospital stay, and mortality [4]. However, improvements in trauma outcomes can be attributed to changes in multiple elements of trauma care, rather than a single magic bullet and they must be tracked and documented over the years before their effects can be reported. To demonstrate what actions are needed by the managers and planners to develop a trauma system, the Trauma System Maturity Index (TSMI) was adopted by the WHO [5]. The four elements of the TSMI include prehospital trauma care, education and training, facility based trauma care, and quality assurance.

The Hamad Trauma Center [HTC] is the only national tertiary trauma care center in Qatar; it is based in the Hamad General Hospital (HGH), the hub of the not-for-profit governmental healthcare system in the country. The evolution of the trauma system in Qatar began in late 2007 with the establishment of the trauma surgery section at HGH as a part of improving emergency care project with the University of Pittsburgh [UOP] [6]. The trauma surgery section was expanding the trauma care team to include trauma surgeons, emergency physicians, nurses, anesthetists, intensivists, clinical pharmacists, a psychologist, physiotherapists, nutritionists and other allied medical personnel. All components of the trauma system in Qatar, from the Ambulance Service to Rehabilitation, rapidly evolved over the years by achieving several important developmental milestones, including international accreditations. The journey from inception to maturation involved several advancements in multiple components of the trauma system such as pre-hospital care (ambulance services), trauma resuscitation unit at the trauma center, development of the Qatar Trauma Registry, trauma performance

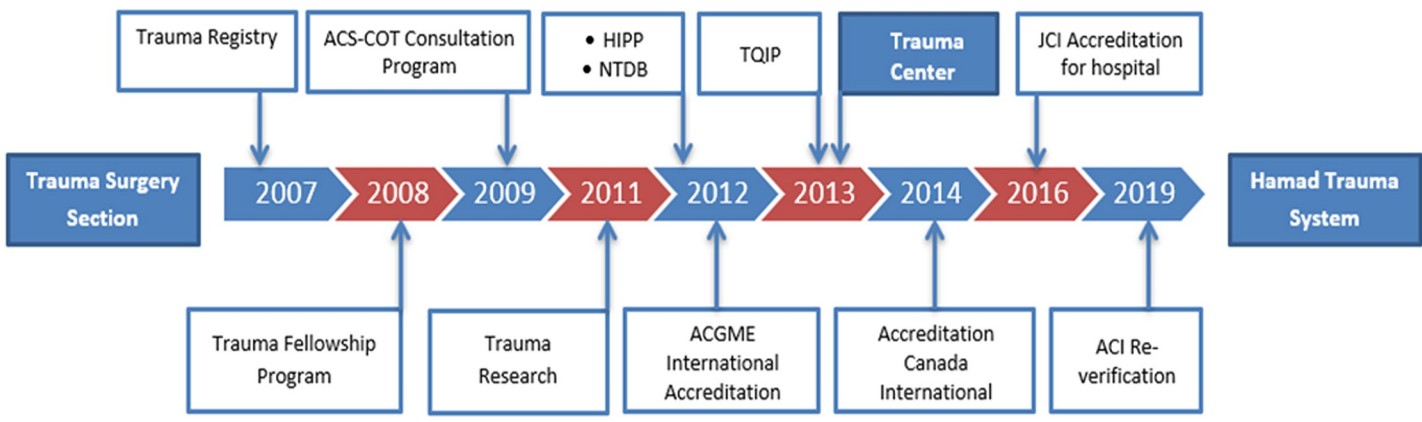

**Fig 1. Milestones in development and maturation of the trauma system in Qatar.**

improvement program (TPIP), Trauma and Critical Care Fellowship Program (TCCFP), Hamad Injury Prevention Program (HIPP), post trauma follow up clinics (including psycho-therapy clinic) and the Clinical Research Unit. The trauma system was accredited by the Accreditation Canada Distinction Program in the year 2014. This was achieved through the creation and implementation of specific processes and demonstration of functionality of essential elements of the trauma system that were required by the accrediting body. It was the process of attaining this accreditation, as the hub of a trauma system, which contributed to the maturation and brought several positive changes in the provision of the trauma care quality [6]. Based on this accreditation, our center was recognized as a level 1 trauma center. The present study aims to determine the effects of this maturation process on the outcomes of trauma patients in a rapidly developing country (Qatar). We hypothesized that the international accreditation is a major step in the maturation process of the national trauma system that is associated with improved patient care and outcomes (Fig 1).

## Methods

A retrospective analysis of data obtained from the Qatar trauma registry at HGH was conducted for trauma patients who were admitted to the tertiary level 1 HTC, from 01/01/2010 to 31/12/2018. All trauma patients, regardless of age, gender, type of mechanism and severity of injury were included whereas patients with incomplete relevant data were excluded. HTC is the only level 1 trauma center in Qatar that provides trauma care, free of charge, for all residents and citizens (a total of 2.7 million inhabitants).

Collected data included admission and discharge details; demographic information (age & gender); injury types (i.e., work-related) and mechanisms (i.e. motor vehicle crash (MVC), pedestrian injuries, fall from heights, fall of heavy objects, sport-related injuries and assault), body regions injured (head, chest, spine or abdomen); Glasgow Coma Score (GCS), Abbreviated Injury Scales (AIS) and Injury Severity Score (ISS); level of trauma activation (i.e., T1); blood alcohol concentration (BAC) screening; radiography; blood transfusion; activation of Massive Transfusion Protocol (MTP); use of intracranial pressure monitors (ICP); exploratory laparotomy; craniotomy or craniectomy; complications (ventilator associated pneumonia (VAP) or sepsis; length of stay (LOS) in the Emergency Department (ED), intensive care unit (ICU) and hospital; discharge to rehabilitation or home; pre-hospital and in-hospital mortality.

The total pre-hospital time (TPT) in this study was defined as the time from the Emergency Medical Service (EMS) notification to hospital arrival. The GCS is a neurological scale that provides reliable and objective evaluation of consciousness; ranging from 3 to 15. The AIS is an anatomically-based injury severity scoring system that classifies each injury by body region on a 6- point scale. The ISS assesses the combined effects of multiply injured patients and is based on the AIS [7, 8]. The ISS ranges from 1 to 75 and correlates with mortality, morbidity and other measures of severity. The level of trauma activation depends on the injury severity and priority for care. T1 or priority 1 (P1) represents critically injured patients who need immediate life-saving interventions. Massive transfusion is defined as replacement of >1 blood volume in 24 hours, or >50% of blood volume in 4 hours. MTP is activated for the critically bleeding patients who are anticipated to require massive transfusion. ED time was categorized into ≤ 4h and >4h. The cut-off 4h was chosen based on the National Hospital Ambulatory Medical Care Survey published by the CDC [9].

Trauma patients included in this study were classified into two groups, before (group 1; from 01/01/2010 to 31/10/2014) and after accreditation in 2014 (group 2; 01/11/2014 to 31/12/2018). The HTC accreditation in 2014 was a landmark achievement, which endorsed the quality and safety of the trauma care provided. HTC became the first trauma center outside Canada to obtain this recognition. The accreditation was awarded to the HTC based on its degree of compliance with the Accreditation Canada standards; achievement of performance indicator thresholds; implementation of trauma protocols, clinical practice guidelines and commitment to excellence and innovation [10].

The study groups were compared in terms of patient characteristics, TPT, prehospital mortality and in-hospital outcomes. Also, the trauma research output of the trauma system was compared based on the number and types of PubMed indexed publications in the 2 eras of the study.

As data were retrieved anonymously with no direct contact with patients, it was not possible to involve patients or the public in the design or conduct of our research. This observational retrospective study received an expedited review and was approved by the Institutional Review Board (HMC IRB# MRC-01-20-103) of Hamad medical corporation, Qatar.

### Statistical analysis

The statistical data analysis was performed using the SPSS 21.0 Statistical Analysis program (SPSS Inc., Chicago, IL). Categorical data were presented as numbers and proportions. Continuous data were reported as mean with the standard deviations (SD) for normally distributed data and as median [interquartile rage (IQR)] for not normally distributed data. While comparing two groups, differences in categorical data were assessed using the Chi-square tests and the significance of statistical differences were attributed to a two tailed p value of <0.05. The continuous data between two groups were compared using Student's T test for normally distributed data and non-parametric tests for the comparison of not normally distributed data. The two study periods were further compared based on the injury severity (ISS and GCS) and ED length of stay. Time series analysis (forecasting) for outcome was performed using expert modeler for total, prehospital and in-hospital mortality.

### Results

A total of 15,864 patients were admitted during the study period, of which 8316 (52%) and 7548 (48%) were admitted during the pre and post—accreditation period, respectively (**Table 1**). The mean age of the study cohort was 31 years and the majority (90%) were males. The most common mechanism of injury was MVC (33%) followed by falls (29%). Nearly 30%

**Table 1. Comparative analysis of the pre-and post-accreditation periods.**

| Variables | Overall (N = 15,864) | Pre-accreditation (n = 8316, 52.4%) | Post-accreditation (n = 7548, 47.6%) | P-value |
|---|---|---|---|---|
| Age | 30.4±16.5 | 29.9±16.4 | 30.8±16.5 | 0.002 |
| Males | 14278 (90.0) | 7465 (89.8) | 6813 (90.3) | 0.29 |
| Mechanism of injury (n = 15,848) | | | | |
| • Motor vehicle crash | 5238 (33.1) | 2683 (32.3) | 2555 (33.9) | 0.09 |
| • Fall | 4589 (28.9) | 2444 (29.4) | 2145 (28.4) | |
| • Pedestrian | 2042 (12.9) | 1061 (12.8) | 981(13.0) | |
| • Fall of heavy objects | 960 (6.1) | 532 (6.4) | 428 (5.7) | |
| • Other | 3019 (19.0) | 1582 (19.1) | 1437 (19.0) | |
| Work related injuries | 4759 (30.0) | 2432 (29.2) | 2327 (30.8) | 0.03 |
| Prehospital time | 55 IQR (37–72) | 53 IQR (37–68) | 59 IQR (39–75) | 0.001 |
| Referrals | | | | |
| • From outside country | 206 (1.3) | 170 (2.0) | 36 (0.5) | 0.001 |
| • Within Qatar | 2102 (13.2) | 984 (11.8) | 1118 (14.8) | |
| BAC screening | 8732 (55.0) | 3648 (43.9) | 5084 (67.4) | 0.001 |
| Alcohol positives (n = 8732) | 1027 (11.8) | 482 (13.2) | 545 (10.7) | 0.001 |
| Head Injury | 4879 (30.8) | 2471 (29.7) | 2408 (31.9) | 0.001 |
| Head CT | 10445 (65.8) | 4944 (59.5) | 5501 (72.9) | 0.001 |
| Chest CT | 8819 (55.6) | 3564 (42.9) | 5255 (69.6) | 0.001 |
| Abdominal CT | 10145 (63.9) | 4702 (56.5) | 5443 (72.1) | 0.001 |
| CT-spine | 10152 (64.0) | 4540 (54.6) | 5612 (74.4) | 0.001 |
| ISS | 10 IQR (5–17) | 9 (5–15) | 10 (5–17) | 0.001 |
| ISS>12 (%) | 6049 (39.5) | 2937 (37.1) | 3112 (42.2) | 0.001 |
| Trauma level1(T1) activation | 2807 (17.7) | 1157 (13.9) | 1650 (21.9) | 0.001 |
| Blood transfusion | 2660 (16.8) | 1236 (14.9) | 1424 (18.9) | 0.001 |
| MTP activation | 588 (3.7) | 247 (3.0) | 341 (4.5) | 0.001 |
| ICP monitor | 362 (2.3) | 147 (1.8) | 215 (2.8) | 0.001 |
| Exploratory Laparotomy | 819 (5.2) | 433 (5.2) | 386 (5.1) | 0.79 |
| Craniotomy/Craniectomy | 463 (2.9) | 221 (2.7) | 242 (3.2) | 0.04 |
| VAP | 656 (4.1) | 406 (4.9) | 249 (3.2) | 0.001 |
| Sepsis | 195 (1.2) | 102 (1.2) | 93 (1.2) | 0.97 |
| ICU stay | 4 IQR (2–11) | 4 IQR (2–11) | 4 IQR (2–10) | 0.65 |
| Hospital LOS | 5 IQR (2–12) | 5 IQR (2–13) | 4 IQR (2–11) | 0.001 |
| ED stay time (min) | 356 IQR (200–598) | 322 IQR (180–525) | 405 IQR (225–690) | 0.001 |
| Discharge to rehabilitation | 844 (5.4) | 350 (4.3) | 494 (6.6) | 0.001 |
| **Total mortality** | 1280 (8.1) | 608 (7.3) | 672 (8.9) | 0.001 |
| • Prehospital mortality | 603 (47.1) | 251 (41.3) | 352 (52.4) | |
| • In-hospital mortality | 677 (52.9) | 357 (58.7) | 320 (47.6) | |
| **Time of in-hospital death** | | | | |
| • <48 hrs. in-hospital | 349 (52.6) | 193 (55.3) | 156 (49.7) | 0.22 |
| • >48 hrs. to 7 days | 183 (27.6) | 95 (27.2) | 88 (25.2) | |
| • >1 week | 131 (19.8) | 61 (17.5) | 70 (20.1) | |

Data expressed as count (percentage) or mean ± standard deviation or median with interquartile range (IQR) whenever appropriate; BAC: Blood Alcohol Concentration; CT: computed tomography; ISS; Injury Severity Score; GCS: Glasgow Coma Score; AIS: Abbreviated Injury Scale; MTP: Massive Transfusion Protocol; ICP: Intracranial pressure monitor; VAP: Ventilator Associated Pneumonia; ICU; Intensive Care Unit; ED: Emergency Department: LOS: Length of Stay

**Table 2. Injury severity and mortality per year.**

| Year | Hospitalization (n) | In-hospital mortality | In-hospital mortality per 100 admissions | Median ISS | ISS Interquartile range | |
|------|---------------------|-----------------------|------------------------------------------|------------|------|------|
| | | | | | 25% | 75% |
| 2010 | 1382 | 66 | 4.8 | 9 | 5 | 17 |
| 2011 | 1697 | 55 | 3.2 | 9 | 5 | 14 |
| 2012 | 1736 | 67 | 3.9 | 9 | 5 | 14 |
| 2013 | 1716 | 76 | 4.4 | 10 | 5 | 17 |
| 2014 | 1604 | 107 | 6.7 | 10 | 5 | 17 |
| 2015 | 1672 | 99 | 5.9 | 10 | 5 | 17 |
| 2016 | 1757 | 79 | 4.5 | 10 | 5 | 17 |
| 2017 | 1695 | 56 | 3.3 | 10 | 5 | 17 |
| 2018 | 1719 | 72 | 4.2 | 10 | 5 | 17 |

Injury severity Score (ISS)

of injuries reported were work-related injuries. Although the male predominance remained unchanged, the mean age was found slightly higher in group 2. There was no statistically significant difference in the mechanism of injuries in both study groups. However, there was a slight increase in work-related injuries observed in group 2.

The median TPT was 55 minutes; this was 6 min longer in group 2 when compared to group 1 (59 vs.53 minutes, p = 0.001). Overall, the BAC was tested in 55% of all patients; and there was a significant increase in the screening rate in group 2 (67% vs. 44%, p = 0.001). The median ED stay time was higher in group 2 (405 vs. 322 minutes).

During the study period, computed tomography (CT) of the head, abdomen, chest and spine were performed in 66%, 64%, 56% and 34% of patients respectively. Head injury was reported in 31% of patients. Utilization of CT increased significantly in group 2 (CT head, chest, abdomen and spine). The ISS was significantly higher in group 2 in which over 40% had mean ISS of greater than12. The GCS at the scene and upon hospital arrival were lower in group 2 (p<0.05). In addition, the proportion of T1 activation was higher in group 2 (22% vs. 14%, p = 0.001). **Table 2** shows the pattern of in-hospital fatality per cases and ISS by year. The ISS trend was in parallel with the mortality trend across the years. **Table 3** shows the breakdown of ISS, GCS and ED stay. Patients with mild injury (ISS 0–9 and GCS 14–15) were more

**Table 3. Pre and post accreditation based on injury severity and emergency (ED) length of stay.**

| | Pre-accreditation | Post-accreditation | P-value |
|------|-------------------|--------------------|---------|
| **Injury severity score (ISS)** | | | 0.001 |
| 0–9 | 3395 (51.5) | 3125 (44.0) | |
| 10–15 | 1494 (22.7 | 1795 (25.3) | |
| 16–24 | 979 (14.9) | 1217 (17.1) | |
| >25 | 720 (10.9) | 962 (13.6) | |
| **Glasgow coma scale** (GCS) | | | 0.001 |
| </ = 3 | 766 (11.6) | 1020 (14.4) | |
| 4–13 | 306 (4.6) | 472 (6.7) | |
| 14–15 | 5556 (83.8) | 5597 (79.0) | |
| **ED length of stay** | | | 0.001 |
| ≤ 4 hr | 2389 (36.2) | 1834 (26.3) | |
| > 4hr | 4202 (63.8) | 5133 (73.7) | |

prevalent in the pre-accreditation period whereas moderate or severe injuries were more likely encountered in the post-accreditation period. In the 1st group, one third of cases stayed 4h or less in the ED in comparison to one quarter of the cases from the 2nd group.

Interventions like blood transfusion; MTP activation; craniotomy or craniectomy and ICP monitor use were higher in group 2 (p<0.05). Exploratory laparotomies were comparable across the groups. Despite the significant increase in diagnostic and treatment interventions including blood transfusions in group 2 suggesting a higher proportion of severely injured patients, complications were less in group 2 (i.e., VAP) or comparable (i.e., sepsis). ICU LOS was comparable, with shorter hospital LOS in group 2 (4 vs. 5 days, p = 0.001). Discharge to rehabilitation services was higher in group 2 than group 1 (7% vs. 4%, p = 0.001).

The overall mortality was 8% in the study cohort. It was significantly higher in group 2 when compared to group 1 (9% vs. 7%, p = 0.001) mostly due to higher pre-hospital mortality in this group (52% vs. 41%, p = 0.001). The in-hospital mortality was significantly lower in group 2 (48% vs. 59%, p = 0.001). There were no significant differences in the number of days in hospital before death across the study groups. Fig 2 shows time series analysis for outcome (mortality). For the total mortality, there was relative increasing trend with cyclical variation (stationary R-squared 0.74, p = 0.09), in-hospital mortality showed a decreasing trend after the year of 2014 (Stationary R-squared 0.76, p = 0.09) and prehospital mortality showed increasing trend (stationary R-squared 0.73, R square 0.29, p = 0.73 and 0 outliers).

## Discussion

The present study describes the effect of implementing the required processes and protocols in order to attain the international recognition and accreditation of the national trauma system

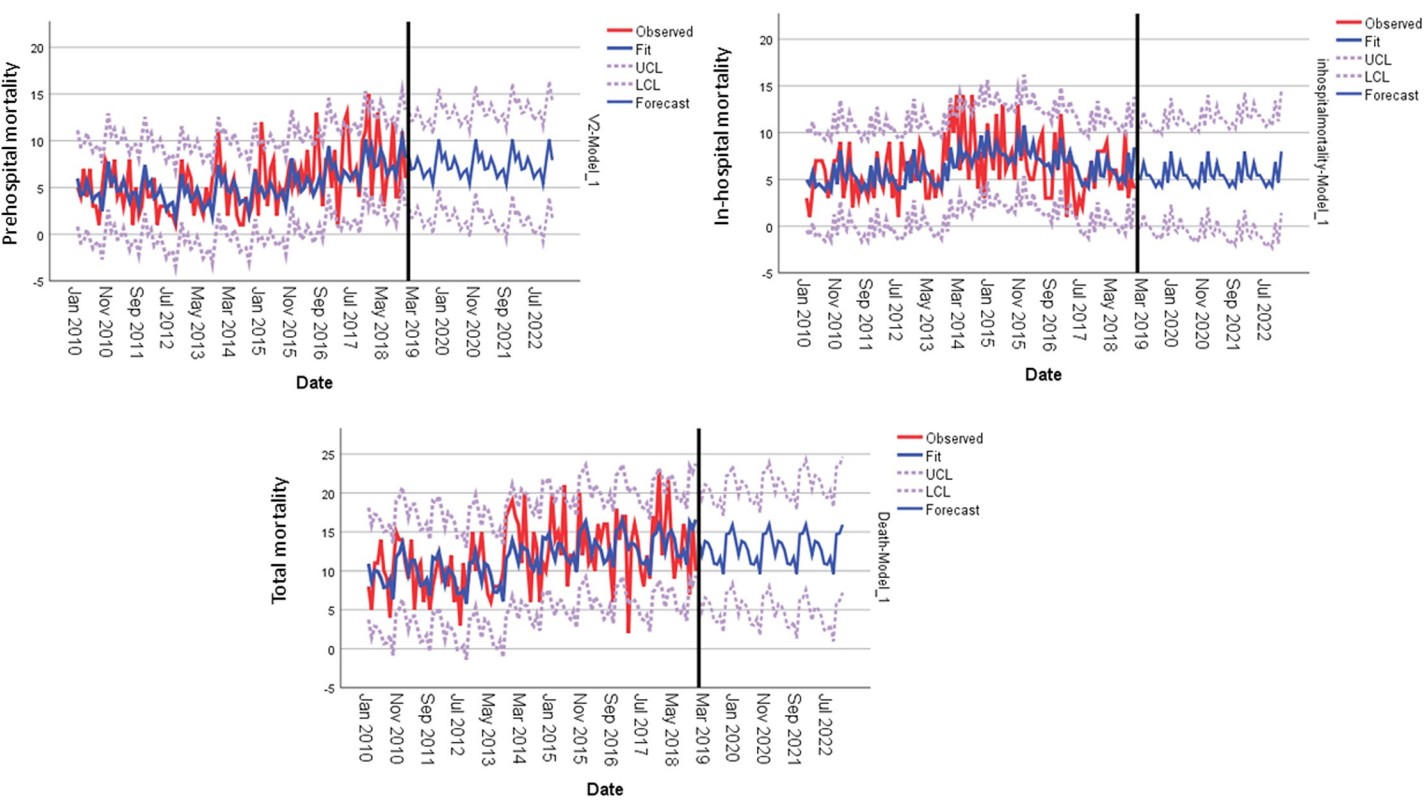

**Fig 2. Time series analysis for outcome.**

in a rapidly developing country. The process of care at the trauma center observed significant changes over the years as evidenced by an increase in the utilization of the diagnostic procedures and treatment/interventions in the post-accreditation period, i.e. after 2014. Furthermore, there were significant declines in the critical complication rates of VAP and total hospital LOS post-accreditation. Despite the higher prehospital mortality after 2014, which can be attributed partly to the increase in the whole country population as well as severity of injuries, there was a significant decrease in the in-hospital mortality. Also, there was an improvement in the compliance for sentinel processes for traumatic brain injury (i.e. ICP monitor) and bleeding patients (MTP activation).

The elements of the national trauma system of Qatar rapidly evolved with the targeted creation of the essential elements of trauma system and center, i.e. Education, Research, Quality Improvement, Registry and Injury Prevention, and ACS Orange Book use. This process, which happened over few years, was guided by the requirements set by Accreditation Canada International and the Committee on Trauma, American College of Surgeons and each was based on the acquisition of material, manpower and financial resources that were combined to fulfill each of these requirements. This process culminated in the achievement of several important developmental milestones, such as the creation of the ACGME-based Trauma and Critical Care Fellowship, increased number of publications from the Trauma Research Unit, establishing the Qatar national trauma registry, and the international accreditations of the trauma center, the laboratory department (CAP) and the ambulance service (JCI).

Provision of trauma care at accredited centers by a dedicated professional team substantially improves the quality of care and outcomes of trauma patients. Peitzman et al., demonstrated that maturation of trauma center was associated with improved survival, fewer complications, and a shorter length of stay among trauma patients [4]. The study included a total of 15,303 trauma patients admitted between 1987 and 1995 in a level 1 trauma center in Pennsylvania of the United States (US). This center was accredited as a level I trauma center by the state in 1987–1988. In this period, dedicated trauma operating room and trauma ICU were established. In addition, dedicated trauma surgeons and full-time emergency physicians were appointed, and the trauma registry was upgraded [4]. The study duration (9 years) and number of patients included in the US study were similar to our study, and the key findings were also comparable in the post-accreditation period.

Harmsen et al., also demonstrated improved trauma care in the Netherlands by reporting significant decline in mortality over the years in a level 1 trauma center [11]. This study included severely injured patients (ISS>15) and the trauma scores across the study groups (2004–2005 vs. 2014) were comparable. However, the number of trauma surgical interventions and the number of blood products transfused were less in the more recent cohort when compared to the earlier cohort. The study concluded that the observed survival was better than the predicted survival in the recent cohort [11]. Our study included all trauma patients regardless of the severity of injuries. Patients admitted in the post-accreditation period had a higher proportion of severe injury, i.e., patients with ISS>12 were greater in the post-accreditation period compared to the pre-accreditation period (42% vs. 37%). Some of the core performance indicators determined by the Accreditation Canada included trauma patients with ISS >12; ED LOS, hospital LOS and trauma mortality [10]. In addition, higher ISS, higher proportion of head injury patients, lower GCS and higher proportion of T1 activations reported in group 2 suggest that these patients had severer injuries and required more surgical interventions and blood transfusions which contrasts to the Harmsen et al., study. In addition to the survival benefits and shorter hospital LOS, Lansink et al., demonstrated shorter ICU LOS associated with trauma center maturation when adjusted for age, ISS, and survival [12].

Barquist et al., demonstrated a significant reduction in mortality rate associated with trauma system maturation. This was partly attributed to positive changes in field triage and early transportation of patients to trauma centers [13]. Takahashi et al. demonstrated that an increase in the rate of transfer of patients to a higher-level medical institution might result in decreased mortality rate [14]. In our study, the median TPT increased in group 2, probably due to the injury characteristics of the victims, i.e. more severe injuries including head injury, and therefore on-scene management might be longer.

The efforts of both the TPIP and HIPP programs may have accounted for the higher rate of compliance to BAC screening (in 2014), MTP activation (started in 2010), trauma activation and field triage criteria. The joint efforts and the development of local standards and protocols such as TBI management in partnership of trauma and neurosurgery may explain the observed increase in ICP monitoring and craniotomy according to the Trauma Brain Foundation guidelines. In parallel, higher compliance with ICU bundles of care such as the ventilator bundle, hand washing and antimicrobial stewardship may account for the lower VAP and complications rates and shorter LOS. Moreover, increased cooperation within the trauma with other disciplines such as rehabilitation resulted in facilitated transfer and higher patient acceptance rates by rehabilitation programs despite the rapid growth in the Qatar population and relatively less beds in these programs. The population growth and stress on the existing hospital beds may account for the increase in the time spent in the ED observed after 2014.

Qatar provides pre-hospital ambulance emergency services (ground and helicopter) to all residents, free of charge, which is integrated with the primary, secondary and tertiary hospital services across the country. The "trauma by-pass protocol" field triage criteria allows for transfer of patients directly from the scene to HTC. The helicopter EMS system in Qatar, is accredited by the European Aeromedical Institute and evolved from a single-patient MD902 helicopter a few years ago operated only daylight hours, to two Augusta-Westland 139 helicopters operating on 24-hour basis [15]. Ambulances are staffed by Critical Care Paramedics (CCP) and Ambulance Paramedics (AP). The CCP program (n = 9) was established in 2008 to provide additional clinical skills and expertise for the management of critically ill patients during transfer. These interventions include diagnostic and therapeutic emergency cardiac procedures, surgical airway techniques, needle thoracentesis, rapid sequence intubation and administration of a comprehensive critical care medication formulary (e.g. tranexamic acid). The addition of the mentioned procedures may explain in part the observed trivial increase in the TPT.

Also, there are many changes in the prehospital notification system that underwent significant improvement from Hamad Communication Center, to the recent addition of state-of-the-art National Communication Command Center. Another addition to consider beside the step up approach system that the EMS (ambulance use: Emergency Medical Technicians (alpha unit), CCP (Charlie unit), the navigation tools after 2014 (i.e.: GPS system that allows the ground ambulance to reach the target location faster) and the country wide distribution of ambulances to allow timely response to the scence. Also, Disaster Planning and program was added in the Trauma System, such as the establishment and creation of the Mass Trauma Incident (MTI) protocol.

Education is one of the components of the trauma system which is closely linked to the performance improvements activities, recording and analyzing quality indicators that are international benchmarking feasible such as NTSD, TQIP program, CAP, Ambulance services, JCI, and Leapforg [16]. A TCCFP was started in 2011. This program utilized an ACGME-based curriculum that focused on augmenting the critical care skills and knowledge of all physicians who were part of the HTC staff, regardless of their primary medical specialty. It was later expanded as a requirement for all physicians who joined the staff. The TCCFP, has provided

education and training to our physicians covering trauma and critical care services, which have been associated with enhanced clinical outcomes as described by Pronovost et al., and others [17, 18]. The Advanced Trauma Life Support course, for both physicians and nurses, is also a required course for all current HTC staff and all must maintain their certification in such standard courses, resulting in a consistent conduct and language of trauma care at the HTC.

Clinical research remains as a key component of a matured trauma system to disseminate experience and contribute to global evidence-based patient care. Although research in emergency settings is challenging due to the time-critical nature of interventions, there was a rapid improvement in trauma research output at HTC as evident by the increasing number of research grants and publications. The trauma research unit was established in mid-2011. Trauma and vascular surgery publications in PubMed indexed journals went from 24 manuscripts before 2014 to 115 by the end of 2018. Raeeszadeh et al., identified Qatar as one of the leading countries in producing peer-reviewed publications in trauma surgery in MEDLINE; where Qatar was the third highest in 2017 [19].

Injury prevention education and outreach program by the HIPP largely contributed to improving public health and safety. Since its establishment, the HIPP obtained various research funds to support injury prevention education and surveillance programs. Road safety programs are fundamental; the HIPP proposed to comply with the global five-pillar matrix plan which includes road safety management, safer roads, safer vehicles, safer road users, and post-crash response. It is believed that trauma mortality associated with RTIs can be reduced by half [6]. However, the increase in population and number of motor vehicles registered in Qatar over the years resulted in elevated risk for RTIs [20, 21]. The number of road motor vehicles per 1,000 inhabitants was 450 in 2010 and increased by 28% in the following 8 years to reach 575 per 1,000 inhabitants in 2018 [20].

## Limitations

The retrospective design of the present study constitutes a significant limitation, particularly on time-related variables. Nearly 28% of TPT data were missing, while other missing data on age, gender, mechanism of injury, ED stay, ISS and GCS were low ranging between 3 and 4%. The study also lacked more detailed pre-hospital time data that could be used to analyze the maturation of pre-hospital care within the trauma system. It is a single center experience; however, there is no other center in our region has awarded this accreditation yet. Although time series analysis showed the trend of mortality, the forecasting analysis showed wide prediction intervals.

## Conclusions

The Qatar Trauma system witnessed a rapid evolution following accreditation of its sole Trauma Center. In-hospital mortality significantly dropped after the Accreditation compared to the previous era while prehospital mortality increased. Accreditation also brought other benefits including a reduction in complications such as VAP and hospital length of stay. The findings corroborate the hypothesis that international accreditation represents an important element in the maturation of the Trauma System. Further studies with prospective design are required to explore the maturation process of all components of the trauma system.

## Acknowledgments

The authors thank all the staff of the trauma registry database at the trauma surgery section.

## Author Contributions

**Conceptualization:** Ayman El-Menyar, Mohammad Asim, Rafael Consunji, Husham Abdelrahman, Ashok Parchani, Hassan Al-Thani.

**Data curation:** Ahammad Mekkodathil, Mohammad Asim, Ruben Peralta, Sandro Rizoli, Husham Abdelrahman, Monira Mollazehi.

**Formal analysis:** Ayman El-Menyar, Ahammad Mekkodathil, Mohammad Asim, Rafael Consunji, Gustav Strandvik, Ruben Peralta, Sandro Rizoli, Monira Mollazehi, Ashok Parchani.

**Methodology:** Ayman El-Menyar, Ahammad Mekkodathil, Mohammad Asim, Rafael Consunji, Gustav Strandvik, Monira Mollazehi, Hassan Al-Thani.

**Writing – original draft:** Ayman El-Menyar, Ahammad Mekkodathil.

**Writing – review & editing:** Rafael Consunji, Gustav Strandvik, Ruben Peralta, Sandro Rizoli, Husham Abdelrahman, Ashok Parchani, Hassan Al-Thani.

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
