## [Decision Letter · Decision Letter 0]

28 Sep 2020

PONE-D-20-26293

Maturation Process and International Accreditation of Trauma System in a Rapidly Developing Country

PLOS ONE

Dear Dr. El-Menyar,

Thank you for submitting your manuscript to PLOS ONE. After careful consideration, we feel that it has merit but does not fully meet PLOS ONE’s publication criteria as it currently stands. Therefore, we invite you to submit a revised version of the manuscript that addresses the points raised during the review process.

We look forward to receiving your revised manuscript.

Kind regards,

Zsolt J. Balogh, MD, PhD, FRACS

Academic Editor

PLOS ONE

Journal Requirements:

3. Thank you for including in your ethics statement:

"This observational retrospective 315 study received an expedited review and was

316 approved by the Institutional Review Board (HMC IRB# MRC-01-20-103).".

i) Please amend your current ethics statement to include the full name of the ethics committee/institutional review board(s) that approved your specific study. 

ii) Once you have amended this/these statement(s) in the Methods section of the manuscript, please add the same text to the “Ethics Statement” field of the submission form (via “Edit Submission”).

Additional Editor Comments (if provided):

Dear Authors,

Our reviewers have major concerns about your methodology and found your paper rather marginal.

Please attempt to address their questions and concerns in an itemised fashion to potentially reconsider it for review.

Reviewers' comments:

Reviewer's Responses to Questions

**Comments to the Author**

1. Is the manuscript technically sound, and do the data support the conclusions?

Reviewer #1: No

Reviewer #2: Yes

2. Has the statistical analysis been performed appropriately and rigorously? 

Reviewer #1: No

Reviewer #2: No

3. Have the authors made all data underlying the findings in their manuscript fully available?

Reviewer #1: No

Reviewer #2: Yes

4. Is the manuscript presented in an intelligible fashion and written in standard English?

Reviewer #1: Yes

Reviewer #2: Yes

5. Review Comments to the Author

Reviewer #1: Thankyou for the opportunity to review the article “Maturation Process and International Accreditation of Trauma System in a Rapidly Developing Country”.

I commend the authors on being leaders in improving trauma care in Qatar.

To determine if the positive results are because of the improved trauma system, there needs to be;

1. Specific detail about the trauma system and the changes introduced

2. A time series analysis of outcomes. It is not possible to do a pre/post analysis based on accreditation, because many of the changes were made leading up to accreditation – so the conclusions are not valid.

You have the data to do this, and it is possible. I suggest this is conducted and the manuscript revised and resubmitted.

I also have some minor comments around the abstract

The background in the abstract is a hypothesis, there needs to be a little bit of introduction / need for the paper provided – for example, some detail about the changes that were introduced as part of the trauma system

Methods: what type of analyses were performed?

In the conclusion you have introduced new results.

Reviewer #2: Thank you for your submission which sought to determine the effects of trauma system maturation to Level 1 trauma care on the outcomes of trauma patients in a rapidly developing country (Qatar).

Abstract: Appropriate representation of the manuscript.

Introduction: Provides thorough information as to the reason for the study.

Patients and methods:

Please add appropriate reference to AIS coding section.

It is difficult to see the split of injury severity within the data. The provision of ISS grouping i.e. <12, 12-15, >15 or similar would provide better understanding of the nuances of care for injured groups and also assist in the understanding of access to time critical intervention. An example of data that requires greater understanding is the ED LOS times. 405 mins is significant for those with ISS >15, but not so with those <15.

Results: Appropriately described.

Discussion: See comments

Comments:

The paragraph on education either needs to be related to the study data of removed as it does not currently add value to the discussion.

This is an important body of work and I would like to commend you on your commitment to improving the care provided to injured people and their families. Development and maturation of systems of care are never easy, particularly in developing countries.

Whilst I am enthusiastic about the results here, I do feel that further breakdown and analysis of data is required (as suggested above). I look forward to future developments.

Thank you for your paper.

6. PLOS authors have the option to publish the peer review history of their article (what does this mean?). If published, this will include your full peer review and any attached files.

Reviewer #1: No

Reviewer #2: No

---

## [Author Response · Author response to Decision Letter 0]

7 Oct 2020

Journal Requirements:

3. Thank you for including in your ethics statement:

"This observational retrospective 315 study received an expedited review and was

316 approved by the Institutional Review Board (HMC IRB# MRC-01-20-103).".

i) Please amend your current ethics statement to include the full name of the ethics committee/institutional review board(s) that approved your specific study.

ii) Once you have amended this/these statement(s) in the Methods section of the manuscript, please add the same text to the “Ethics Statement” field of the submission form (via “Edit Submission”).

Reply: done

Comments to the Author

5. Review Comments to the Author

Reviewer #1: Thank you for the opportunity to review the article “Maturation Process and International Accreditation of Trauma System in a Rapidly Developing Country”.

I commend the authors on being leaders in improving trauma care in Qatar.

To determine if the positive results are because of the improved trauma system, there needs to be;

1. Specific detail about the trauma system and the changes introduced

Reply: thanks, detailed trauma system changes were given in the introduction and discussion in addition to figure 1. See line 81-101 and 216-226

2. A time series analysis of outcomes. It is not possible to do a pre/post analysis based on accreditation, because many of the changes were made leading up to accreditation – so the conclusions are not valid.

You have the data to do this, and it is possible. I suggest this is conducted and the manuscript revised and resubmitted.

Reply: time series analysis done and shown in figure 2

I also have some minor comments around the abstract

The background in the abstract is a hypothesis, there needs to be a little bit of introduction / need for the paper provided – for example, some detail about the changes that were introduced as part of the trauma system

Reply: done in line 30-33

Methods: what type of analyses were performed?

Reply: analyses methods added

In the conclusion you have introduced new results.

Reply: conclusion revised.

Reviewer #2: Thank you for your submission which sought to determine the effects of trauma system maturation to Level 1 trauma care on the outcomes of trauma patients in a rapidly developing country (Qatar).

Abstract: Appropriate representation of the manuscript.

Introduction: Provides thorough information as to the reason for the study.

Patients and methods:

Please add appropriate reference to AIS coding section.

Reply: done , ref 7 and 8 added

It is difficult to see the split of injury severity within the data. The provision of ISS grouping i.e. <12, 12-15, >15 or similar would provide better understanding of the nuances of care for injured groups and also assist in the understanding of access to time critical intervention. An example of data that requires greater understanding is the ED LOS times. 405 mins is significant for those with ISS >15, but not so with those <15.

Reply: new table (3) added with the breakdown of injury severity , GCS and ED time

Results: Appropriately described.

Discussion: See comments

Comments:

The paragraph on education either needs to be related to the study data of removed as it does not currently add value to the discussion

Reply: education paragraph revised.

This is an important body of work and I would like to commend you on your commitment to improving the care provided to injured people and their families. Development and maturation of systems of care are never easy, particularly in developing countries.

Whilst I am enthusiastic about the results here, I do feel that further breakdown and analysis of data is required (as suggested above). I look forward to future developments.

Reply: further analyses done and shown in table 3 and figure 2

---

## [Decision Letter · Decision Letter 1]

25 Nov 2020

Maturation Process and International Accreditation of Trauma System in a Rapidly Developing Country

PONE-D-20-26293R1

Dear Dr. El-Menyar,

We’re pleased to inform you that your manuscript has been judged scientifically suitable for publication and will be formally accepted for publication once it meets all outstanding technical requirements.

Kind regards,

Zsolt J. Balogh, MD, PhD, FRACS

Academic Editor

PLOS ONE

Additional Editor Comments (optional):

Reviewers' comments:

Reviewer's Responses to Questions

**Comments to the Author**

1. If the authors have adequately addressed your comments raised in a previous round of review and you feel that this manuscript is now acceptable for publication, you may indicate that here to bypass the “Comments to the Author” section, enter your conflict of interest statement in the “Confidential to Editor” section, and submit your "Accept" recommendation.

Reviewer #1: All comments have been addressed

2. Is the manuscript technically sound, and do the data support the conclusions?

Reviewer #1: Yes

3. Has the statistical analysis been performed appropriately and rigorously? 

Reviewer #1: Yes

4. Have the authors made all data underlying the findings in their manuscript fully available?

Reviewer #1: No

5. Is the manuscript presented in an intelligible fashion and written in standard English?

Reviewer #1: Yes

6. Review Comments to the Author

Reviewer #1: Thankyou for addressing my comments. I have no further comments

........................................................

7. PLOS authors have the option to publish the peer review history of their article (what does this mean?). If published, this will include your full peer review and any attached files.

Reviewer #1: No

---

## [Editor Report · Acceptance letter]

2 Dec 2020

PONE-D-20-26293R1 

Maturation Process and International Accreditation of Trauma System in a Rapidly Developing Country 

Dear Dr. El-Menyar:

I'm pleased to inform you that your manuscript has been deemed suitable for publication in PLOS ONE. Congratulations! Your manuscript is now with our production department. 

Kind regards, 

on behalf of

Dr. Zsolt J. Balogh 

Academic Editor

PLOS ONE